# Transcriptome Analyses of Barley Roots Inoculated with Novel *Paenibacillus* sp. and *Erwinia gerundensis* Strains Reveal Beneficial Early-Stage Plant–Bacteria Interactions

**DOI:** 10.3390/plants10091802

**Published:** 2021-08-30

**Authors:** Tongda Li, Ross Mann, Jatinder Kaur, German Spangenberg, Timothy Sawbridge

**Affiliations:** 1Agriculture Victoria, AgriBio, Centre for AgriBioscience, Bundoora, VIC 3083, Australia; ross.mann@agriculture.vic.gov.au (R.M.); jatinder.kaur@agriculture.vic.gov.au (J.K.); german.spangenberg@agriculture.vic.gov.au (G.S.); tim.sawbridge@agriculture.vic.gov.au (T.S.); 2School of Applied Systems Biology, La Trobe University, Bundoora, VIC 3083, Australia

**Keywords:** *Paenibacillus*, *Erwinia gerundensis*, barley, growth-promotion, interaction, RNA-seq

## Abstract

Plant growth-promoting bacteria can improve host plant traits including nutrient uptake and metabolism and tolerance to biotic and abiotic stresses. Understanding the molecular basis of plant–bacteria interactions using dual RNA-seq analyses provides key knowledge of both host and bacteria simultaneously, leading to future enhancements of beneficial interactions. In this study, dual RNA-seq analyses were performed to provide insights into the early-stage interactions between barley seedlings and three novel bacterial strains (two *Paenibacillus* sp. strains and one *Erwinia gerundensis* strain) isolated from the perennial ryegrass seed microbiome. Differentially expressed bacterial and barley genes/transcripts involved in plant–bacteria interactions were identified, with varying species- and strain-specific responses. Overall, transcriptome profiles suggested that all three strains improved stress response, signal transduction, and nutrient uptake and metabolism of barley seedlings. Results also suggested potential improvements in seedling root growth via repressing ethylene biosynthesis in roots. Bacterial secondary metabolite gene clusters producing compounds that are potentially associated with interactions with the barley endophytic microbiome and associated with stress tolerance of plants under nutrient limiting conditions were also identified. The results of this study provided the molecular basis of plant growth-promoting activities of three novel bacterial strains in barley, laid a solid foundation for the future development of these three bacterial strains as biofertilisers, and identified key differences between bacterial strains of the same species in their responses to plants.

## 1. Introduction

Plants and bacteria can establish mutualistic beneficial interactions or undesirable pathogenic interactions [1], leading to great impacts on the performance of agriculturally important crops and pastures. Plant growth-promoting (PGP) bacteria possess genes conferring beneficial traits to their host plants, and can act as biofertilisers and bioprotectants, leading to significant increases in yield and improved tolerance to both biotic and abiotic stresses in plants [2,3]. *Paenibacillus polymyxa* strains have long been described as PGP bacteria that can improve the nutrient uptake and metabolism of plants via biological nitrogen fixation and phytohormone production, and protect plants from phytopathogens via synthesising bioactive secondary metabolites [4,5,6]. *Erwinia gerundensis* is a newly identified species that was isolated from pome fruit trees and is associated with multiple plant hosts across different continents [7]. Understanding the interactions between plants and bacteria, especially PGP bacteria, has the potential to improve the overall performance of agricultural plants.

The beneficial interactions between legumes and rhizobia have been extensively studied, revealing the molecular basis and regulatory pathways of each stage of their interaction [8,9]. However, such knowledge remains to be discovered for many other PGP bacteria and non-leguminous plants. An ideal method to study the molecular basis of plant–bacteria interactions is dual RNA-seq analysis, which can provide transcriptome profiles of both host plants and bacteria simultaneously [10]. While dual RNA-seq analysis has been widely used to reveal the interactions between plants and phytopathogens [11,12,13], its application in studying the interactions between plants and PGP bacteria is limited. Camilios-Neto et al. [14] reported the first case of using dual RNA-seq analyses to demonstrate the interactions between wheat and a PGP bacterium *Azospirillum brasilense*, revealing improvements in plant nutrient acquisition and metabolism. Recent work by Liu et al. [15] demonstrated that interactions between *P. polymyxa* YC0136 and tobacco plants enhanced phytohormone transduction and systemic resistance against pathogens in the plant, as well as stimulated auxin biosynthesis in the bacterial strain. Such promising results suggest that dual RNA-seq analyses should be used to deepen our understandings of interactions between more plant species, especially agricultural crops, and other novel PGP bacteria including *Paenibacillus* spp. and *E. gerundensis*.

In this study, we utilised two novel *Paenibacillus* sp. strains (S02 and S25) and one novel *E. gerundensis* strain (AR) isolated from the perennial ryegrass (*Lolium perenne* L. cv. Alto) microbiome [16,17]. Preliminary characterisation showed that the two *Paenibacillus* sp. strains were genetically closely related to *P. polymyxa* and had strong bioprotection and biological nitrogen fixation activities in vitro [17], and the *E. gerundensis* strain was able to grow in low nitrogen conditions in vitro and enhance plant root development [18], making them ideal candidates for further characterisation. An early-stage plant–bacteria interaction assay was conducted using barley seedlings and the three strains. Barley seedlings and bacterial strains were co-incubated for six hours and harvested for RNA extraction. Dual RNA-seq analyses were then performed to identify differentially expressed genes/transcripts associated with the early-stage plant–bacteria interaction and to provide insights into the molecular basis of the interaction, with focuses on (1) initial contact between the bacteria and plant, (2) bacterial plant growth-promoting genes, plant nutrient uptake and metabolism, and (3) bacterial secondary metabolites.

## 2. Results

### 2.1. Transcriptome Sequencing—An Overview

An early-stage plant–bacteria interaction assay was conducted using barley seedlings and the three strains (S02 and S25: *Paenibacillus* sp.; AR: *E. gerundensis*). Seedlings and bacterial strains were co-incubated in Nutrient Broth (NB; all three strains) or Burk’s N-free medium (S02 only). Plant root tissues and the bacterial culture were separated after six hours of co-incubation and harvested for RNA extraction. A transcriptome sequencing experiment was designed to explore genes/transcripts that were associated with the interaction. A 150 bp paired-end library prepared from cDNA from samples used in the assay generated an average of 61.7 million clean reads per bacterial sample and 132.4 million clean reads per plant sample (Appendix A). Transcript quantification showed that 80–90% and 85–90% of reads from bacterial and plant samples were mapped to the corresponding transcriptome reference, including the transcriptome generated from Prokka [19] annotation for bacterial samples and a barley reference transcriptome BaRTv1.0 [20] for plant samples, respectively. Biological variability was checked by comparing the normalised counts of mapped reads within the biological replicates generated by differential gene expression (DGE) analyses using Pearson correlation coefficients. All biological replicates had a correlation coefficient of 0.92–0.99, except the root samples of barley co-incubated with *Paenibacillus* sp. strain S02 in Burk’s N-free medium (correlation coefficient: 0.84–0.98), suggesting high data reproducibility of this study and the robust nature of the methodology.

### 2.2. Transcriptome Analyses—An Overview

DGE analyses clearly demonstrated changes in transcriptome profiles caused by plant–bacteria interactions. For bacterial samples, the biological replicates of all three strains formed distinctive clusters along the PC1 axis based on the presence/absence of barley seedlings (Figure 1). For plant samples, four distinctive clusters containing the biological replicates of each treatment (the presence/absence of bacteria in different media) were identified (Figure 2). Seedlings co-incubated with strain AR (*E. gerundensis*) were separated from seedlings co-incubated with strain S02 and S25 (*Paenibacillus* sp.) along the PC1 axis. Moreover, seedlings co-incubated with strain S02 in Burk’s N-free medium were separated from other seedlings co-incubated with bacterial strains in NB along the PC2 axis. These results suggested that the transcriptome profiles of seedlings were affected by both the bacterial species/strain they were co-incubated with and the medium used in the assay. Moreover, when comparing seedlings co-incubated with bacterial strains in NB, seedlings co-incubated with strain AR or S25 formed distinct clusters that were separated from the control seedlings along all three axes (PC1–PC3, Figure 3). Conversely, seedlings co-incubated with strain S02 formed a cluster with the control seedlings along axes PC1 and PC2, only separating along the PC3 axis (Figure 3, right) which accounted for 4.29% of the total variances. These results suggested seedlings co-incubated with strain S02 produced transcriptome profiles similar to the control seedlings, unlike strain AR and strain S25 which have triggered more obvious changes in transcriptome profiles of barley seedlings.

DGE analyses successfully identified genes that were differentially expressed caused by plant–bacteria interactions (Table 1). For bacteria, the DGE analyses compared transcriptome profiles of bacteria when barley seedlings were present and absent. When NB was used, strain AR (*E. gerundensis*) had 4009 genes that passed the abundance filter, 1380 of which were differentially expressed when seedlings were present. For *Paenibacillus* sp. strains S25 and S02, 5013 and 5266 genes passed the abundance filter, respectively, and 2945 and 2890 genes were differentially expressed when seedlings were present, respectively. Moreover, strain S02 cultured in Burk’s N-free medium had 5032 genes that passed the abundance filter and 2524 genes that were differentially expressed when seedlings were present. Interestingly, strain-specific responses were identified from the two *Paenibacillus* sp. strains in NB (Figure 4) even though the two strains are genetically highly similar (average nucleotide identity = 97.78%) and share 4332 conserved genes [17]. Amongst 4332 conserved genes, there were 997 genes that were only differentially expressed by strain S02 and 1104 genes that were only differentially expressed by strain S25. There were also 1317 genes that were differentially expressed by both strains, including 228 genes that were induced in strain S02 but repressed in strain S25 and another 228 genes that were repressed in strain S02 but induced in strain S25. There were also 490 genes that were upregulated in both strains and 371 genes that were downregulated in both strains, and 914 genes that were not differentially expressed by either strain.

For barley, the DGE analyses compared transcriptome profiles of seedlings with and without the inoculated bacterial strains. Barley seedlings co-incubated with strains AR, S25 and S02 in NB had 37,073, 35,365 and 34,798 genes that passed the abundance filter respectively, and 13,948, 13,648 and 9129 genes that were differentially expressed when bacterial strains were present. When Burk’s N-free medium was used, seedlings co-incubated with strain S02 had 31,502 genes that passed the abundance filter and 10,806 genes that were differentially expressed when the strain was present. Overall, 22,015 barley genes were differentially expressed during the plant–bacteria interaction assay using NB, including 3862 genes that were shared by interactions with all three strains, and 5117, 4020 and 2030 genes that were unique to interactions with strain AR, S25 and S02, respectively (Figure 5). GO enrichment analysis using 3862 differentially expressed barley genes shared by all three strains identified an overrepresented (*p* < 0.05) GO category associated with sequence-specific DNA binding (GO:0043565), suggesting the transcriptional regulation of plant–bacteria interactions. There were no overrepresented GO categories associated with disease responses and plant defence mechanisms detected using those barley genes. GO enrichment analysis of the 8,067 differentially expressed barley genes that were only associated with the two *Paenibacillus* sp. strains (S02 and S25) revealed overrepresented (*p* < 0.05) GO categories associated with nitrogen metabolism, including nitrogen compound transport (GO:0015112) and organonitrogen compound metabolic process (GO:1901564). Moreover, compared with seedlings inoculated with *Paenibacillus* sp. strain S02, seedlings inoculated with *Paenibacillus* sp. strain S25 shared more differentially expressed genes with seedlings inoculated with *E. gerundensis* strain AR. GO enrichment analysis of the 7611 genes shared by seedlings inoculated with strain S25 and AR revealed overrepresented (*p* < 0.05) GO categories associated with stress responses (GO:0006950, 0006979).

### 2.3. Transcriptome Analyses—Functional Genes Associated with Plant—Bacteria Interaction

The DGE analyses clearly demonstrated that the transcriptome profiles of bacterial strains and roots of barley seedlings were shaped by the interactions between them, causing significant changes in expression levels of some genes. Specific genes that may be involved in plant–bacteria interaction are described and discussed below. Gene expression levels (or transcripts expression levels for plant data) when bacteria/plants were present were represented as approximate fold-changes in relation to the expression levels when bacteria/plants were absent unless otherwise specified.

#### 2.3.1. Bacterial Initial Contact with Plants

Bacterial genes that are involved in the initial contact with plants (chemotaxis and biofilm formation) were differentially expressed by all three strains (AR, S02 and S25) in NB (Appendix A). The expressions of methyl-accepting chemotaxis proteins, which are the predominant chemoreceptors that sense the presence of signal molecules and nutrients produced by plants as root exudates [21], were downregulated (up to a 3-fold decrease) in strain AR but were upregulated (up to a 3.9-fold increase) in strain S02 and S25. In addition, functional annotation also identified other chemotaxis proteins that were downregulated in strain AR and S25 but were upregulated in strain S02. The flagellar motor switch proteins, which are utilised by bacteria to move towards favourable environments [22], were highly expressed by strain AR and S25 (up to a 5.01-fold increase) but not by strain S02. Moreover, transporter proteins for sugars, which are the major content of root exudates [23], were upregulated in all three strains (up to a 2.17-fold increase).

Biofilm formation was described as an adaptive strategy used by bacteria to enable successful host colonisation [24]. In this study, the *Paenibacillus* sp. strain S02 even formed visible biofilms on the root surface within three hours of co-incubation with barley seedlings (Figure 6). DGE analyses showed that genes that are involved in biofilm formation, including the biosynthesis of exopolysaccharide, glycogen and cellulose, were highly expressed by strain AR and S25 (up to a 17.61-fold increase) but not by strain S02 (Appendix A). However, comparisons of the control (i.e., when the barley seedlings were absent) of the two *Paenibacillus* sp. strains (S02 and S25) suggested that those genes were actively expressed by strain S02 with up to a 39.39-fold increase (Appendix A).

#### 2.3.2. Plant Growth-Promoting Genes

The two *Paenibacillus* sp. strains used in this study (S02 and S25) were characterised by our laboratory, revealing the presence of a comprehensive set of plant growth-promoting genes including biological nitrogen fixation, inorganic and organic phosphate solubilisation and transportation, as well as phytohormone (indole-3-acetic acid) production and transportation [17]. DGE analyses showed that most of these genes were either not differentially expressed or downregulated in expression levels when barley seedlings were present (Appendix A). However, the expression of one of the auxin efflux carriers genes was upregulated with a 2.34- and 986.13-fold increase for strain S02 and S25, respectively. Furthermore, it is known that biological nitrogen fixation of *P. polymyxa* strains requires low environmental nitrogen content [25], therefore a plant–bacteria interaction assay using Burk’s N-free medium was conducted for *Paenibacillus* sp. strain S02. It was shown that strain S02 carries a highly active nitrogen fixation (*nif*) operon when growing in Burk’s N-free medium [17]. DGE analyses showed that the expression levels of the *nif* operon were upregulated with up to an 11.12-fold increase when barley seedlings were present (Appendix A). Moreover, the transporting and binding proteins of molybdenum, which is an essential part of the nitrogenase [26], were also highly expressed (up to a 7.22-fold increase, Appendix A), suggesting that they were co-induced with the *nif* operon under low N conditions.

Compared to the two *Paenibacillus* sp. strains (S02 and S25), the *E. gerundensis* strain AR carries a reduced set of plant growth-promoting genes (phosphate transporters and auxin efflux carriers). However, similar to the two *Paenibacillus* sp. strains, these genes were either not differentially expressed or downregulated in expression levels when barley seedlings were present (Appendix A).

#### 2.3.3. Secondary Metabolite Biosynthesis Gene Clusters

Previous analyses conducted by our laboratory suggested that the three strains used in this study (AR, S02 and S25) possess multiple secondary metabolite biosynthesis gene clusters [17,18], and DGE analyses showed that their expression levels were regulated by plant–bacteria interactions. For the two *Paenibacillus* sp. strains (Appendix A), the expression levels of the core biosynthetic genes of secondary metabolite gene clusters encoding known antimicrobial compounds polymyxin (C15) [27] and tridecaptin (C10) [28] were upregulated in strain S02 but were downregulated in strain S25. The expression level of the core biosynthetic genes of another antimicrobial compound paenilan (C7) [29] was not changed in strain S02 but was increased by 2.73-fold in strain S25. Both strains carry an antifungal compound fusaricidin cluster (C1) [30] whose core biosynthetic gene was downregulated in expression. Strain S25 also carries a unique lanthipeptide cluster with the core biosynthetic genes being downregulated. As for the secondary metabolite gene clusters that encode novel products, whilst the expression levels of the core biosynthetic genes were either not changed or downregulated, the expression levels of the core biosynthetic genes encoding a novel non-ribosomal peptide (C11) were increased by up to 2.53-fold in strain S02 and up to 13.60-fold in strain S25. The expression levels of a siderophore cluster (C2) were also upregulated in both *Paenibacillus* sp. strains.

Interestingly, the core biosynthetic genes of most of the secondary metabolite gene clusters were highly expressed by strain S02 in Burk’s N-free medium when barley seedlings were present, including all clusters encoding known antimicrobial compounds with up to a 12.29-fold increase and clusters encoding novel compounds with up to a 269.16-fold increase (Appendix A).

The *E. gerundensis* strain AR carries a carotenoid biosynthesis cluster that was upregulated when barley seedlings were present (up to a 2.37-fold increase in expression levels). The expression levels of the core biosynthetic genes of the remaining six secondary metabolite gene clusters were either not changed or downregulated (Appendix A).

#### 2.3.4. Defence and Stress Response Mechanisms Utilised by Barley Seedlings

DGE analyses revealed differentially expressed barley transcripts associated with plant defence and stress response mechanisms. The expressions of defence-related proteins, including disease resistance proteins and heat shock proteins [31], were regulated by all three strains, however, the *Paenibacillus* sp. strain S02 induced less of those proteins that were differentially expressed when compared to the other *Paenibacillus* sp. strain S25 and the *E. gerundensis* strain AR (Appendix A). Moreover, transcripts encoding the R-gene-coded resistance protein leucine-rich repeat receptor kinase [32] were only upregulated by strain AR (a 1.75-fold increase, Appendix A). Similarly, whilst transcripts encoding inhibitors of bacterial degradative enzymes such as the polygalacturonase [33] and xylanase [34] were differentially expressed with all three strains, only strain AR upregulated the expression levels (Appendix A). Furthermore, strain AR also induced the increased expressions of more transcripts encoding endoglucanases, which are released by plants to degrade the cell wall of pathogens [35], when compared with strains S02 and S25 (Appendix A).

The expressions of stress response-related proteins were also regulated by all three strains (Appendix A). The *E. gerundensis* strain AR induced the increased expressions of more transcripts encoding caffeoyl CoA *O*-methyltransferase, a key enzyme involved in the biosynthesis of lignin that supports the mechanical strength of plant cells [36] when compared to the two *Paenibacillus* sp. strains (S02 and S25). The same trend was also observed for the stress response proteins glutamate decarboxylase [37] and ubiquitin-activating enzyme E1 [38] when using NB as the medium. However, strain S02 induced more differentially expressed transcripts encoding glutamate decarboxylase when using Burk’s N-free medium. Furthermore, only strain AR induced differentially expressions of transcripts encoding ascorbate peroxidase, an enzyme that scavenges reactive oxygen species (ROS) released by plants under environmental stress [39].

#### 2.3.5. Differentially Expressed Barley Transcripts Associated with Signal Transduction and Ethylene Biosynthesis

DGE analyses revealed differentially expressed barley transcripts associated with plant signal transduction (Appendix A). The expression of transcripts encoding GTP binding proteins and ADP-ribosylation factors, which are involved in plant cellular processes by controlling and relaying signals [40,41], were upregulated by all three strains, especially the *E. gerundensis* strain AR.

DGE analyses also revealed differentially expressed barley transcripts associated with ethylene biosynthesis (Appendix A). To synthesise the phytohormone ethylene, plants first use ACC synthase to convert S-adenosyl-L-methionine into 1-aminocyclopropane-1-carboxylic acid, which is then converted into ethylene by ACC oxidase [42]. The expressions of transcripts encoding ACC oxidase were greatly suppressed by all three strains (up to a 314.69-fold decrease).

#### 2.3.6. Differentially Expressed Barley Transcripts Associated with Nutrient Uptake and Metabolism

DGE analyses revealed differentially expressed barley transcripts associated with nutrient uptake and metabolism. Expressions of transcripts encoding high-affinity transporters associated with nitrate, iron, potassium, sulphate and inorganic phosphate were regulated (Appendix A). All three strains greatly repressed the expression of transcripts encoding high-affinity sulfate transporters (up to a 493.77-fold decrease). A similar repression of expression was also identified from transcripts encoding high-affinity nitrate transporters (up to a 478.53-fold decrease), however, strain AR still induced the expression of three transcripts encoding the protein (a 128.40-fold increase). Expressions of transcripts encoding high-affinity potassium transporters were induced by both strain AR and S25 (up to a 37.21-fold increase) but were repressed by strain S02 (a 1.72-fold decrease). Moreover, only strains AR and S02 upregulated the expression of transcripts encoding high-affinity iron transporters, and only the two *Paenibacillus* sp. strains downregulated the expression of transcripts encoding high-affinity inorganic phosphate transporters.

Expressions of transcripts associated with nitrogen transport and metabolism were also regulated (Appendix A). Compared to the two *Paenibacillus* sp. strains S02 and S25, the *E. gerundensis* strain AR caused differential expressions of more transcripts encoding ammonium transporters, including one with a 160.67-fold increase and another with a 457.99-fold decrease. Expressions of transcripts encoding glutamine synthetase, which is the principal enzyme involved in nitrogen assimilation [43], and transcripts encoding aspartate aminotransferase, which plays an important role in nitrogen metabolism [44], were upregulated by all three strains. Interestingly, only strain AR greatly induced the expression of transcripts encoding anthocyanidin-*O*-glucosyltransferase involved in anthocyanin biosynthesis (up to a 1063.42-fold increase), which was reported to be induced by low nitrogen stress [45].

Expressions of transcripts associated with carbohydrate metabolism were also regulated (Appendix A). The *E. gerundensis* strain AR greatly induced the expression of transcripts encoding UDP-glucose pyrophosphorylase (up to a 753.55-fold increase), which is involved in the biosynthesis of sucrose and the plant cell wall [46]. All three strains also induced the expression of transcripts encoding sucrose synthase (up to a 30.36-fold increase), which is the key enzyme of cellulose synthesis [47].

## 3. Discussion

### 3.1. Dual RNA-seq Analyses of Bacterial Strains and Barley Seedling Roots

This is the first study that utilised dual RNA-seq analyses to investigate the early-stage interactions between barley and *E. gerundensis* and *P. polymyxa* strains. *E. gerundensis* is a newly identified species [7] and there has been no research focusing on the transcriptome of the bacterial strain or the inoculated plants as yet. There are a few published studies of the transcriptome or proteome of plants inoculated with *P. polymyxa* strains [48,49,50], but only Liu et al. [15] reported changes in the bacterial transcriptome associated with plant–bacteria interactions. By sequencing the transcriptome of both the host barley seedlings and the inoculated bacterial strains, this study was able to identify over 20,000 barley genes and over 2800 bacterial genes that were differentially expressed caused by the plant–bacteria interaction. It provided a comprehensive transcriptome profile of both the bacteria and the plants that could be examined to understand the molecular basis of plant–bacteria interactions, especially between PGP bacteria and agriculturally important crops and pastures. Future dual RNA-seq studies are required to deepen our understanding of such interactions and to potentially contribute to the development of bacterial biofertilisers and improved breeds of plants.

The quantification of expressed genes/transcripts of RNA-seq analysis relies on high-quality transcriptome references. While acquiring high-quality bacterial transcriptome references has become easier due to recent advances in sequencing technologies [51,52], the availability of high-quality plant transcriptome references is still limited [53]. One of the limitations of the first dual RNA-seq research between plant and PGP bacteria was the lack of a high-quality wheat transcriptome reference [14]. In this study, a high-quality barley transcriptome reference described by Rapazote-Flores et al. [20] was used, leading to improved mapping rates in transcripts quantification (~90%) when compared to the previous transcriptome reference (high-confidence gene set only: ~55%; high- and low-confidence gene sets combined: ~75–80%) described by Mascher et al. [54]. Further research is required to release and improve the reference transcriptomes/genomes of all agriculturally important crops and pastures that underpin RNA-seq analysis studies.

### 3.2. Initial Contact Between Bacteria and Plants—Chemotaxis, Biofilm Formation and Plants’ Defence and Stress Response

Plants produce root exudates containing rich nutrients, which were reported as a key determinant of the composition of the rhizosphere microbiome [55,56]. Bacteria sense the presence of root exudates via chemoreceptors and move towards plants via chemotaxis [57,58], thus creating initial contact between the microbes and plants. In this study, the two *Paenibacillus* sp. strains (S02 and S25) showed higher activities in expressing chemotaxis proteins than the *E. gerundensis* strain (AR) induced by barley seedlings. However, barley seedlings induced active expression of sugar transporters of all three strains and the motor switch proteins of strain AR and S25. While *P. polymyxa* strains are known to colonise the rhizosphere and root tissues [59], *E. gerundensis* was initially described as a cosmopolitan epiphyte of various plants [7]. Hence, it is possible that the *E. gerundensis* strain AR is less capable of sensing root exudates when compared to the two *Paenibacillus* sp. strains S02 and S25 but is still able to move towards and utilise root exudates.

Biofilm formation was described to be important for successful interactions between plants and PGP bacteria [60]. *P. polymyxa* strains are known to form biofilms on plant roots within two hours under gnotobiotic systems and after seven days under soil systems [61], but such information is still missing for *E. gerundensis* strains. In this study, the *Paenibacillus* sp. strain S02 actively expressed genes associated with biofilm formation even when barley seedlings were absent and formed visible biofilms on the root surface in just three hours when barley seedlings were present. The rapid formation of biofilms could also saturate the capacity of the root surface for future colonisation, which could possibly explain the absence of increased expressions of motor switch proteins induced by barley seedlings. For the other *Paenibacillus* sp. strain (S25) and the *E. gerundensis* strain (AR), although no visible biofilm formation was observed, the induced expression of corresponding genes by barley seedlings suggested that the biofilm formation was still ongoing when materials were harvested. Further experiments are required to confirm the presence of such biofilms formed by those two strains using microscopic techniques.

Upon successful colonisation on the root surface, some bacteria are capable of entering plant internal tissues via naturally occurring cracks such as root tips or via active production of plant cell wall degrading enzymes [62,63], thus creating stress conditions to plants. A previous research has shown that *P. polymyxa* strains invade plant roots [61]. The *E. gerundensis* strain AR was isolated from seedlings grown from surface-sterilised seeds of perennial ryegrass [16], suggesting its ability to enter plant tissues. In this study, strain S02 was more active in biofilm formation and root colonisation but induced the expressions of less resistance proteins and stress proteins in barley seedlings when compared to strains S25 and AR, which was consistent with the GO enrichment analysis. Such results suggested that strain S02 is more adaptive to barley, leading to enhanced interactions with hosts and alleviated stresses in hosts. Conversely, inoculating strain AR induced the expressions of more resistance proteins and stress proteins in barley seedlings when compared to strain S02 and S25. Moreover, it also induced the expressions of inhibitors of plant cell wall degrading enzymes (such as xylanase), as well as the enzyme supporting the strength of plane cell wall (caffeoyl CoA *O*-methyltransferase). Such results suggested that the *E. gerundensis* strain AR created more intense stress conditions in barley seedlings when compared to the two *Paenibacillus* sp. strains S02 and S25. Furthermore, strain AR also triggered the expressions of plant defence and stress response genes such as endoglucanase [64] and ascorbate peroxidase [65], suggesting this strain could improve the plant responses to fungal phytopathogens and excessive oxidative stress.

### 3.3. Interactions between Bacteria and Plants—Plant Growth-Promoting Genes, Plant Nutrient Uptake and Metabolism, Signal Transduction and Ethylene Biosynthesis

Bacterial plant growth-promoting genes are a key contributor to the beneficial plant–bacteria interactions. In this study, the expression of biological nitrogen fixation genes (the *nif* operon) of *Paenibacillus* sp. strain S02 was greatly enhanced by barley seedlings under low nitrogen conditions (Burk’s N-free medium), making the strain a promising candidate to be developed as a biofertiliser. Interestingly, the presence of barley seedlings did not enhance the expressions of other plant growth-promoting genes of the three strains. This could be explained by the relatively short period of time of plant–bacteria interactions (six hours). Further research is required to track the expression of those genes in long-term interactions between plants and the three strains.

The beneficial plant–bacteria interactions can greatly improve the nutrient uptake and metabolism of plants [66]. Expressions of barley transcripts encoding high-affinity transporters, which are utilised by plants under low environmental nutrient concentrations [67,68], were downregulated by all three strains for sulphate and nitrate. The *Paenibacillus* sp. strain S02 also repressed the expression of high-affinity potassium transporters. Such results may indicate increased concentrations of these nutrients in the medium released by the action of the bacteria. Transcripts of barley genes associated with nitrogen metabolism and carbohydrate metabolism were observed to be increased when the bacterial strains were present, indicating increased core metabolic activity in the plant. Overall, these results indicated the improved nutrient availability and metabolism in barley seedlings when the three strains were present. Counterintuitively, whilst the nutrient availability was improved, expressions of most of the nutrient transporters in barley seedlings were not induced by the three strains. For instance, expressions of most ammonium transporters were repressed when the two diazotrophic *Paenibacillus* sp. strains (S02 and S05) were present. However, such downregulations were reasonable since excess ammonium can be cytotoxic [43,69,70]. Species-specific responses were also observed in this study. The two *Paenibacillus* sp. strains (S02 and S25) downregulated the expression of barley high-affinity inorganic phosphate transporters, indicating the improved bioavailability of phosphate to seedlings. Such improvement could be explained by the plant growth-promoting genes associated with phosphate solubilisation and transportation carried by the two strains. Besides, the *E. gerundensis* strain AR induced the expression of barley anthocyanidin-*O*-glucosyltransferase. Anthocyanidins are flavonoid pigments that are associated with plants growing under nitrogen-deficient environments [71,72,73]. The induced expressions of the anthocyanidin biosynthesis gene and the GO enrichment analysis of differentially expressed barley genes triggered by the two *Paenibacillus* sp. strains suggested that the *E. gerundensis* strain AR is less capable of improving nitrogen bioavailability and metabolism in barley seedlings when compared to the *Paenibacillus* sp. strains S02 and S25.

In addition to nutrient uptake and metabolism, inoculating the three strains promoted the expressions of plant signal transduction proteins including GTP binding proteins and ADP-ribosylation factors. These proteins were reported to be related to plants’ responses to abiotic and biotic stresses [40,74]. Thus, the induced expressions of these proteins may increase the resilience of inoculated seedlings to stresses. Furthermore, all three strains repressed the ethylene biosynthesis protein ACC oxidase in barley seedlings. Ethylene suppresses root cell elongation and lateral root development [75,76]. Camilios-Neto et al. [14] inoculated wheat with *Azospirillum brasilense* and identified a 3.1-fold decrease in expressions of ACC oxidase and up to 30% increases in root mass in three days. Preliminary glasshouse experiment results have shown that the root length of five-day-old barley seedlings inoculated with the *E. gerundensis* strain AR was 21.8% longer when compared to the uninoculated control [18]. Hence, we proposed that the root growth-promoting activity exhibited by strain AR is analogous, and the two *Paenibacillus* sp. strains (S02 and S25) are also capable of promoting the root growth of barley based on the transcriptome data. Future research (e.g., *in planta* inoculation) is required to validate such root growth-promoting activities of all three strains on more crops such as wheat.

### 3.4. Bacterial Secondary Metabolite

Bacteria produce a wide range of secondary metabolites that have important biological and ecological functions [77,78]. This study demonstrated that the expressions of some bacterial secondary metabolites are potentially associated with plant–bacteria interactions. The *E. gerundensis* strain AR has a carotenoid biosynthesis cluster, and the expressions of the core biosynthetic gene of the cluster were induced 2.37-fold by barley seedlings. Carotenoids play an important role in both pigmentation and cell signalling [79]. Bible et al. [80] reported that a PGP bacterium strain, *Pantoea* sp. YR343, became defective in biofilm formation and root colonisation when its carotenoid biosynthesis was impaired. Given the close phylogenetic relationships between *Erwinia* spp. and *Pantoea* spp. [81], we proposed a hypothesis that carotenoid biosynthesis is also necessary for root colonisation by strain AR. Further targeted research, e.g., creating mutants of the carotenoid biosynthesis cluster, is required to corroborate this hypothesis.

*P. polymyxa* strains are known to produce a diverse range of secondary metabolites, including ones associated with bioprotection and others associated with novel functions [4,5]. The two *Paenibacillus* sp. strains used in this study (S02 and S25) carry multiple secondary metabolite gene clusters encoding known antimicrobial compounds, whose expressions were induced by barley seedlings despite some strain-specific variations. Such increased expression could be associated with the endophytic bacteria of barley seedlings. Although the barley seeds used in this study were surface-sterilised, only epiphytic bacteria would be killed. Barley is known to host a complex endophytic bacterial community [82]. Moreover, the presence of unmapped reads in the bacterial transcriptome sequencing samples also suggested the presence of other bacterial strains. Hence, we postulated that the induced expressions of those antimicrobial secondary metabolites of the two *Paenibacillus* sp. strains were caused by the interactions between these inoculated strains and the endophytic bacterial strains in barley seedlings. To validate this hypothesis, sterile seedlings that are free of both epiphytic and endophytic microbes must be used to repeat the experiment.

Interestingly, the expression of secondary metabolites encoding novel compounds was either not changed or downregulated by barley seedlings when NB was used but was highly increased by barley seedlings when Burk’s N-free medium was used. Compared with NB, Burk’s N-free medium contains fewer nutrients and represents a stress environment to barley seedlings. It was previously suggested that secondary metabolites produced by PGP bacteria can enhance the tolerance to abiotic and biotic stress of plants [83,84]. Therefore, we submitted that these novel secondary metabolites are associated with the stress tolerance response of plants. It is notable that given the complexity of biosynthesis of secondary metabolites, this study focused on the expression levels of core biosynthetic genes of secondary metabolite gene clusters to represent the expression of those metabolites. Future research is required to prove this hypothesis, including purifying, quantifying and characterising the novel compounds with mass spectrometry, in conjunction with *in planta* glasshouse experiments.

### 3.5. Specific Responses Associated with Bacterial Strains and Bacteria/Plant Species, and Their Implications for Enhanced Plant—Bacteria Interactions

Plants recruit and interact with beneficial bacteria by using complex signalling and comprehensive genetic and metabolic controls [3]. As discussed above, all three strains used in this study were isolated from the same host plant species (*L. perenne*) but the two *Paenibacillus* sp. strains exhibited strain-specific behaviours when interacting with barley and caused varied beneficial responses. There were also clear differences between *E. gerundensis* and these strains. Such strain- and species-specific responses of barley transcriptome included different genes associated with stress responses, nutrient uptake and metabolism and phytohormone biosynthesis. Furthermore, strain- and species-specific responses were also proven since different bacterial strains triggered the expression of different transcripts (isoforms) of genes in barley (e.g., ammonium transporter, Appendix A). Whilst such varying responses are anticipated in bacterial strains belonging to different species, it is interesting to observe such varying responses from genetically closely related strains (e.g., *Paenibacillus* sp. strain S02 and S25). Since other Australian *E. gerundensis* strains are available [7,16], further experiments should be conducted to characterise the interactions between those strains and barley seedlings to identify potential strain-specific responses within this species, comparing responses to our *E. gerundensis* strain AR.

Liu et al. [15] recently reported the first dual RNA-seq analysis of interactions between *P. polymyxa* YC0136 and tobacco (*Nicotiana tabacum* L.). *P. polymyxa* YC0136 is closely related to the two *Paenibacillus* sp. strains used in this study, with an average nucleotide identity of 97.81% and 99.29% when compared to strain S02 and S25, respectively [17]. The results from that study are largely consistent with those presented here. Differences in the results presented include fewer differentially expressed bacterial genes (187 vs. 1380 and 2945 in this study) and upregulated expression of the bacterial *pst* genes associated with phosphate transportation. These differences are likely associated with different methodologies used by the two studies, including differing ages of plants and length of co-incubation time. Another reason for the lower number of differentially expressed bacterial genes may be the host plant used. *P. polymyxa* YC0136 was isolated from the tobacco rhizosphere [85], whereas the bacteria described in this study were isolated from perennial ryegrass and tested on a different species, namely barley. Thus *P. polymyxa* YC0136 may be more adapted to the host tested. It remains unclear if this strain would interact with other plant species in a similar pattern when compared to its interactions with the original host plant species. Similarly, will the three strains used in this study interact with their original host (perennial ryegrass) in a similar pattern when compared to their interactions with barley? Will they exhibit a universal pattern of interactions with plants or a host-specific pattern? Further research is required to answer these questions by repeating the experiment using other *P. polymyxa* strains on a wider range of host plant species.

Whilst barley seedlings used in this study were only inoculated with a single bacterial strain, it was demonstrated that co-inoculation of multiple bacterial strains in plants may have synergistic effects [86,87,88]. The microbiome of perennial ryegrass was profiled recently by our laboratory, revealing a complex community of microbes [16]. Given the fact that strain-specific responses were observed in this study, a key question that should orient future studies is how these microbes interact with plants collectively as a community. As the three strains used in this study (S02, S25 and AR) have shown that each has its own role in promoting the growth of barley seedlings, we postulated that a consortium of S02, S25 and AR would be of maximum benefit to the growth of barley seedlings. To affirm this, dual RNA-seq analyses would need to be performed with barley seedlings in the presence of all three strains, with associated differential time points to tease out the interactions as time progresses. Such knowledge is required to characterise these plant-associated bacteria, understand various strain-specific and host-specific responses, and eventually enhance the beneficial interactions to increase the productivity of plants.

## 4. Conclusions

Dual RNA-seq analyses of barley roots co-incubated with the three novel PGP bacterial strains (S02. S25 and AR) revealed remarkable beneficial plant–bacteria interactions. Overall, inoculating PGP bacteria induced gene expression associated with improved stress responses, signal transduction and nutrient uptake and metabolism of barley seedlings. Additionally, expression of barley genes associated with ethylene biosynthesis was greatly suppressed by the three strains, leading to potential improvements in root growth. Expression of bacterial secondary metabolite gene clusters producing both known antimicrobial and novel compounds was also regulated by the plant–bacteria interaction, which are likely correlated with plant stress tolerance. Varying transcriptomic responses of barley seedlings were observed when inoculating strains of different species as well as closely related strains of a species.

This study contributes to our understanding of the molecular basis of interactions between barley and the three novel PGP bacterial strains. The comprehensive transcriptome profiles of both barley seedlings and PGP bacterial strains revealed by this study, especially the species-/strain-specific variations, and the intriguing, regulated expression of genes with unknown functions, warrant further investigation of plant–bacteria interactions using dual RNA-seq analyses. Finally, the beneficial plant–bacteria interactions demonstrated by this study attest to the future use of applied PGP bacteria as substitutes for synthetic chemicals to increase agriculture sustainability.

## 5. Materials and Methods

### 5.1. Assay Design

An assay was designed to examine the transcriptional response in early-stage plant–bacteria interactions. Barley (*Hordeum vulgare* cv. Hindmarsh) seeds and three bacterial strains isolated from the perennial ryegrass (*L. perenne* L. cv. Alto) microbiome were used in this study [16,17], including two novel *Paenibacillus* sp. strains (S02 and S25) and one novel *E. gerundensis* strain (AR). Two media were utilised as the substrates for the assay: a standard medium (Nutrient Broth) and a nitrogen-free medium (Burk’s).

Barley seeds were surface-sterilised (80% ethanol for three minutes, followed by 3 × sterile dH_2_O washes, each one minute) and germinated under sterile conditions (on moistened sterile filter paper in a sealed Petri dish). Bacterial strains were cultured in Nutrient Broth (NB, BD Bioscience) overnight (OD_600_ = 1.0) and cultures were diluted using fresh NB to OD_600_ = 0.7 (final volume = 50 mL). Seedlings (five days old) had their roots submerged in the bacterial culture and were incubated at 26 ℃ for six hours with shaking (100 rpm). Moreover, preliminary characterisations showed that *Paenibacillus* sp. strain S02 is able to actively fix atmospheric nitrogen when growing in Burk’s N-free medium (MgSO_4_, 0.2 g/L; K_2_HPO_4_, 0.8 g/L; KH_2_PO_4_, 0.2 g/L; CaSO_4_, 0.13 g/L; FeCl_3_, 0.00145 g/L; Na_2_MoO_4_, 0.000253 g/L; sucrose, 20 g/L). Hence one extra assay was prepared for strain S02 as described above using Burk’s N-free medium to replace NB. For the blank control (seedlings), seedlings had their roots submerged in sterile media (either NB or Burk’s N-free medium). For the blank control (bacteria), bacteria were cultured without the presence of a seedling. Three samples were prepared as biological replicates for each treatment and control. Plant root tissues were separated from the bacterial culture after six hours of co-incubation. Bacterial cultures were centrifuged to collect pellets. Plant roots and bacterial pellets were used for RNA extraction.

### 5.2. Transcriptome Sequencing

Total RNA was extracted using a TRIzol^™^ Plus RNA Purification Kit (12183555, Thermo Fisher Scientific, Waltham, MA, USA). On-column treatments were conducted using a PureLink^™^ DNase Kit (12185010, Thermo Fisher Scientific) to ensure the complete removal of genomic DNA. RNA samples were assessed for quality (RIN, the RNA integrity number ≥ 7) on an Agilent 2200 TapeStation (Agilent Technologies, Santa Clara, CA, USA). For bacterial RNA samples, ribosomal RNA (rRNA) was depleted using a NEBNext^®^ rRNA Depletion Kit (E7860L, NEB, Ipswich, MA, USA). For plant RNA samples, messenger RNA (mRNA) was enriched using a NEBNext^®^ Poly(A) mRNA Magnetic Isolation Module (E7490L, NEB). Directional RNA-seq libraries were prepared using a NEBNext^®^ Ultra^™^ II Directional RNA Library Prep Kit (E7765, NEB) and sequenced on an Illumina NovaSeq 6000 platform.

### 5.3. Transcriptome Analyses

RNA-seq data (raw reads) were assessed for quality and filtered to remove any adapter and index sequence using fastp [89] with the following parameters: *-w 8 -3 -5 --detect_adapter_for_pe*. Salmon [90] was used to quantify transcripts using the filtered RNA-seq reads with the following parameters: *-l A --validateMappings --numBootstraps 1000 --seqBias*. Complete circular genome sequences were generated and annotated for all three bacterial strains using the methods described in Li et al. [91] (Table 2), which were then used as references for transcript quantification. For the plant samples, a published high-quality barley reference transcript dataset (BaRTv1.0) was used, containing 60,444 genes with 177,240 transcripts [20]. A total of 1000 rounds of bootstraps were performed during transcript quantification to minimise the impact of technical variations. Differential gene expression (DGE) analyses were conducted using an R package sleuth [92]. Likelihood ratio tests were conducted to detect the presence of any significant difference (*q*-value < 0.05) in abundances of each transcript between treatments, and Wald tests were conducted to determine an approximation of the fold-change in abundances of each transcript between treatments. Transcripts that were of ultra-low abundance (defined by having less than 20 mapped reads or were only present in less than three samples) were removed prior to DGE analyses. The differentially expressed transcripts were defined to be significant at *q*-value < 0.05 and absolute fold-change ≥ 1.5. Gene Ontology (GO) enrichment analyses were conducted using g:Profiler [93]. Venn diagrams were generated using BioVenn [94].

## Figures and Tables

**Figure 1 plants-10-01802-f001:**
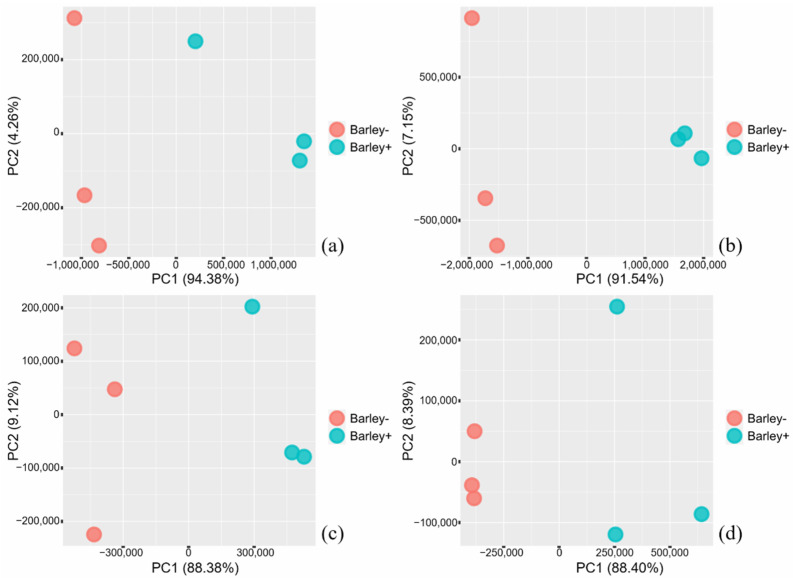
Principal component analysis (PCA) plots representing the clustering of biological replicates based on gene expression levels of (**a**) strain AR (*E. gerundensis*) in Nutrient Broth, (**b**) S25 (*Paenibacillus* sp.) in Nutrient Broth, (**c**) S02 (*Paenibacillus* sp.) in Nutrient Broth, and (**d**) S02 (*Paenibacillus* sp.) in Burk’s N-free medium. Percentage variance explained by each axis is given in brackets. Distinctive clusters that represented the presence (Barley+)/absence (Barley−) of seedlings formed along the PC1 axis, demonstrating the changes in transcriptome profiles caused by the plant–bacteria interactions.

**Figure 2 plants-10-01802-f002:**
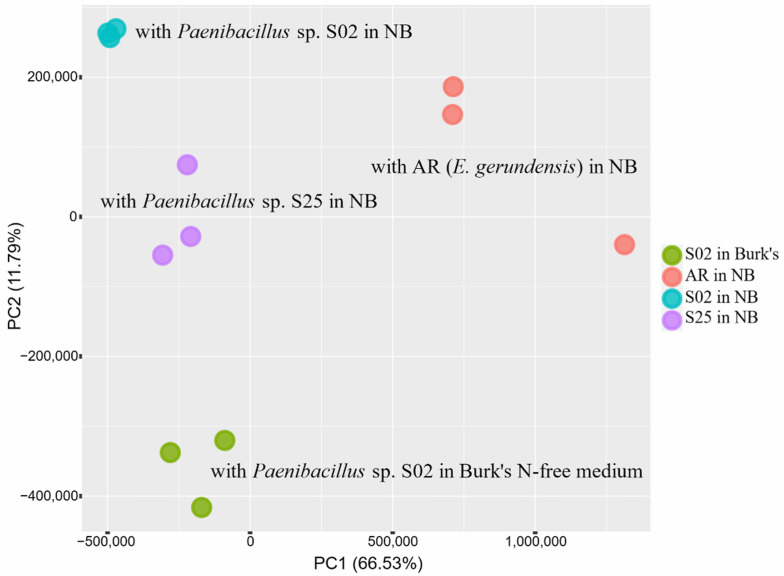
Principal component analysis (PCA) plot representing the clustering of biological replicates based on gene expression levels of barley seedling roots. Percentage variance explained by each axis is given in brackets. Seedlings co-incubated with strain AR (*E. gerundensis*) are separated from the seedlings co-incubated with strain S02 and S25 (*Paenibacillus* sp.) along the PC1 axis, suggesting the effects of two different bacterial species. Seedlings co-incubated with strain S02 in Burk’s N-free medium are separated from the seedlings co-incubated with bacterial strains in Nutrient Broth along the PC2 axis, suggesting the effects of two different media. Seedlings co-incubated with strains S02 and S25 in Nutrient Broth are also separated, suggesting the effects of different strains of the same species.

**Figure 3 plants-10-01802-f003:**
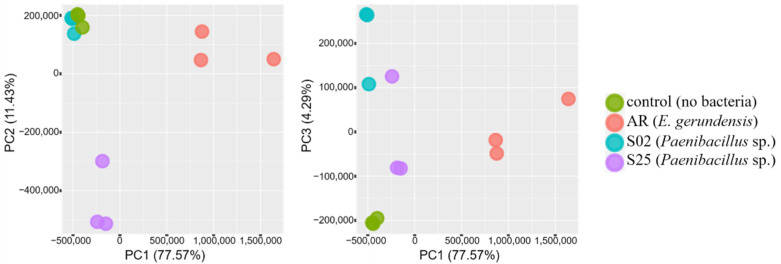
Principal component analysis (PCA) plots representing the clustering of biological replicates based on gene expression levels of barley seedling roots when using Nutrient Broth. Percentage variance explained by each axis is given in brackets. Seedlings co-incubated with strain AR (*E. gerundensis*) or S25 (*Paenibacillus* sp.) are separated from the control seedlings along all three axes (PC1–PC3). However, seedlings co-incubated with strain S02 (*Paenibacillus* sp.) are separated from the control seedlings only along the PC3 axis, which accounts for 4.29% of the total variances, suggesting strains AR and S25 triggered more obvious changes in transcriptome profiles of barley seedlings when compared with strain S02.

**Figure 4 plants-10-01802-f004:**
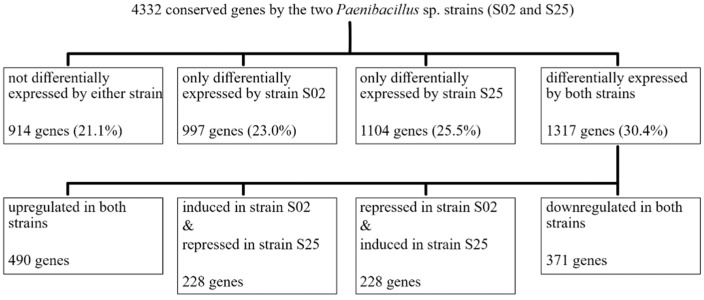
Regulated expressions of 4332 conserved genes of the two *Paenibacillus* sp. strains (S02 and S25) when co-incubated with barley seedlings in Nutrient Broth. Number of genes and the corresponding percentage of total conserved genes are shown for each category. Despite being genetically closely related (average nucleotide identity = 97.78%), the two strains showed strain-specific responses when interacting with barley seedlings.

**Figure 5 plants-10-01802-f005:**
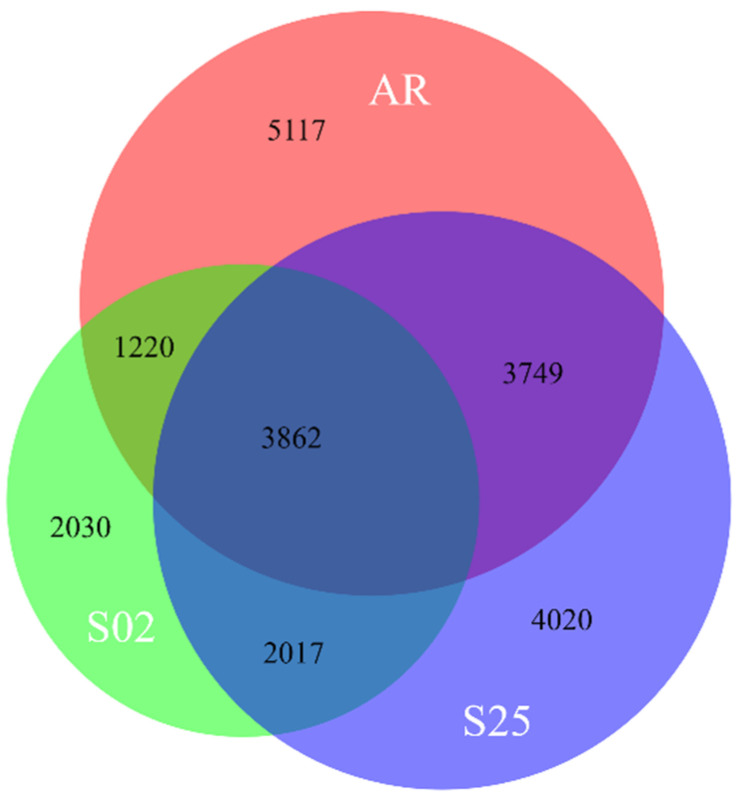
A Venn diagram that shows the number of barley genes that were differentially expressed in roots during the plant–bacteria interaction assay for all three strains in Nutrient Broth. A total of 22,015 genes were differentially expressed. AR: Novel *E. gerundensis* strain; S02/S25: Novel *Paenibacillus* sp. strains.

**Figure 6 plants-10-01802-f006:**
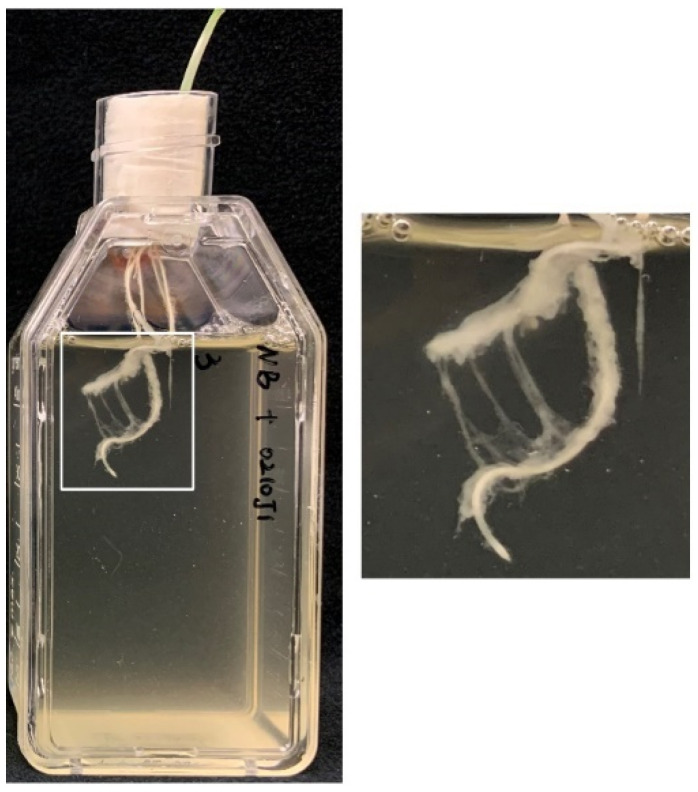
A representative image showing the visible biofilm formed by the *Paenibacillus* sp. strain S02 on the root surface of barley seedlings after three hours of co-incubation.

**Table 1 plants-10-01802-t001:** Bacterial and plant genes that passed the abundance filter and were differentially expressed identified by DGE analyses.

	Sample	Treatment	Medium	No. of Genes Passed the Abundance Filter	No. of Differentially Expressed Genes
Bacteria	AR	Barley seedling	Nutrient Broth	4009	1380
	S25			5013	2945
	S02			5266	2890
			Burk’s N-free	5032	2524
Plant	Barley seedling	AR	Nutrient Broth	37,073	13,948
		S25		35,365	13,648
		S02		34,798	9129
			Burk’s N-free	31,502	10,806

AR: Novel *E. gerundensis* strain; S02/S25: Novel *Paenibacillus* sp. strains.

**Table 2 plants-10-01802-t002:** General genomic characteristics of the three bacterial strains used in the assay.

Strain ID	Genome Size (bp)	No. of Genes
S02 (*Paenibacillus* sp.)	6,060,529	5436
S25 (*Paenibacillus* sp.)	5,958,851	5306
AR (*Erwinia gerundensis*)	4,437,426	4091

## Data Availability

Annotated genome sequences of strains used in this study were deposited in the NCBI GenBank with the accession number PRJNA720480 (AR) and PRJNA720481 (S02 and S25). The results of transcript quantification (raw reads count) were provided as Appendix A.

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
