# Peer review of "Transcriptome Analyses of Barley Roots Inoculated with Novel Paenibacillus sp. and Erwinia gerundensis Strains Reveal Beneficial Early-Stage Plant–Bacteria Interactions"

_plants, 2021, doi:10.3390/plants10091802_

Round 1

Reviewer 1 Report

Tongda et al. have done good research. The quality of the manuscript can be improved further:

  1. Please italicize the scientific names throughout the manuscript.
  2. Some of the parts of the Result section may be appropriately moved to the Materials and Methods section.
  3. Necessary to define the uncommon abbreviations when they first appear in an article.
  4. Better to use full forms in Table and Figure titles.
  5. The authors are recommended to add a separate Conclusion section.

The authors may refer to the attached manuscript for its improvement.

Author Response

Thank you so much for reviewing my manuscript and for your valuable comments. Please see the attachments for my response.

Reviewer 2 Report

The overall quality of the manuscript and  work is, in my opinion, excellent.  The idea to study the transcriptome of both barley and symbiotic bacteria under interaction with each other is quite novel and gives significant insights in the molecular plant microbe interactions. Very good experimental design and methodology, and the description of materials and methods is impressively detailed and accurate. The only drawback is that the gene expression was studied only at 6 hpi. I only have some really minor corrections to suggest

line84-85 and 91--> which figure/table shows the numbers mentioned in the 1st paragraph of the results? I didn't find them in the manuscript

Results

It would be nice to add a PCA plot showing the differentiation of S02 in NB and Burk's medium

in line 190 Is the number 8,067 right? check again Fig5

Although it is impressive, I would move Fig6 to supplementary, since it doesn't really add something to the results and the DEGs involved in biofilm formation were not highly expressed by S02+barley

line 256: Mention the whole word for nif operon (nitrogen fixation) and add the abbreviation in parenthesis (nif).

line 321: differential not differentially. Also you don't specify, up or down?

I think that some of the tables with DEGs (supplementary) could be modified (with more columns) in order to reduce the rows. This way the annotations wouldn't have to be repeated and the extra rows wouldn't take that much space.

Materials and methods

In paragraph 4.1 please mention also here the reference for the 3 strains used in the study

line 620: it's clear from the results that S02 was tested in both liquid media, just clarify it here too. For example "Hence another assay..." could be changed to "One extra assay..."

line 640-642--> I would move this phrase to paragraph 4.3

Author Response

(The authors gave the same response as above.)

Reviewer 3 Report

Comments and Suggestions for Authors

The study generally has a good experimental design with no major flaws that I can see. Manuscript is good prepared and good written.

  • The paper in its entirety, especially references should be adapted to the author's instructions and template of the Plants journal.

For example:

  • Lines 3 and 14 add p to Paenibacillus sp. (Paenibacillus spp)
  • Please follow the comments in the Pdf version in the following lines
  • Lines 38 and 50
  • Lines 77,86,87 and 91
  • Lines 100 1nd 101
  • Lines 230, 236 and 241
  • Lines 230, 236 and 241
  • Lines 245, 255 , 264 and 273
  • Lines 286, 292, 302, 303 and 314
  • Lines 356, 357, 367 and 378
  • Line 483
  • Line 590
  • Line 645
  • All references
  • Please see the attached pdf file for improving the manuscript.

Best regards

Author Response

(The authors gave the same response as above.)
